# Advances in Metabolic Engineering of Plant Monoterpene Indole Alkaloids

**DOI:** 10.3390/biology12081056

**Published:** 2023-07-27

**Authors:** Vonny Salim, Sara-Alexis Jarecki, Marshall Vick, Ryan Miller

**Affiliations:** 1Department of Biological Sciences, Louisiana State University Shreveport, Shreveport, LA 71115, USA; jareckis90@lsus.edu (S.-A.J.); vickm24@lsus.edu (M.V.); 2School of Medicine, Louisiana State University Health New Orleans, New Orleans, LA 70112, USA; rmil17@lsuhsc.edu

**Keywords:** natural product, monoterpene indole alkaloid, metabolic engineering

## Abstract

**Simple Summary:**

Plants produce very diverse chemicals that are used for the treatment of human ailments. However, plants tend to produce these high-value compounds in small amounts. The rapid development of modern synthetic biology tools has facilitated the engineering of living organisms to allow them to manufacture plant natural products. Herein, we highlight the advances of strategies to engineer more robust systems for increased production of plant-based chemicals with medicinal value in non-native systems, including in yeast and tobacco plants. Success in engineering these platforms will ultimately provide more access to plant-based medicines.

**Abstract:**

Monoterpene indole alkaloids (MIAs) encompass a diverse family of over 3000 plant natural products with a wide range of medical applications. Further utilizations of these compounds, however, are hampered due to low levels of abundance in their natural sources, causing difficult isolation and complex multi-steps in uneconomical chemical syntheses. Metabolic engineering of MIA biosynthesis in heterologous hosts is attractive, particularly for increasing the yield of natural products of interest and expanding their chemical diversity. Here, we review recent advances and strategies which have been adopted to engineer microbial and plant systems for the purpose of generating MIAs and discuss the current issues and future developments of manufacturing MIAs by synthetic biology approaches.

## 1. Introduction

Monoterpene indole alkaloids (MIAs) are highly diverse natural products found in a wide variety of medicinal plants, mostly in Gentianales [1,2]. More than 3000 MIAs have been identified to have medicinal properties and are used to treat heart conditions, pain, neurological diseases, microbial infections, malaria, diabetes, and cancer [3,4]. Despite their remarkable pharmaceutical activities, accessing certain quantities of targeted compounds is challenging, particularly due to their low abundance in plants, the difficulty of plant propagation, and the tedious steps of purifying the bioactive compounds from chemically complex extracts. Although the synthetic chemistry of alkaloids is considerably improving, the synthesis processes are lengthy for industrial production, often with impractical separation phases. The most well-known MIAs, vinblastine and vincristine, which have been effectively used in chemotherapies, are harvested from the leaves of the medicinal plant *Catharanthus roseus* (Madagascar periwinkle) [5]. The concentration of these medicinally important compounds is as low as 0.0005% dry weight [6,7,8]. Subsequently, since vinblastine and vincristine are drugs with shortages according to the US Food and Drug Administration in 2019–2020, relying on extractions from *C. roseus* leaves is unsustainable [9,10]. Their low yields and extensive production steps have prompted rigorous efforts to genetically engineer tractable heterologous hosts, including microbial and plant systems for higher titers of these valuable MIAs.

The large-scale sequencing and access to the transcriptomes and genomes of medicinal plants have facilitated the discovery of all genes that are involved in vinblastine biosynthesis in *C. roseus* [11,12,13,14,15]. In recent years, the identification of these previously unknown genes has led to growing interests in manufacturing plant alkaloids in microbial cell factories, and this further demonstrates the benefits of sustainability with low cost. The selection of microbial hosts has also expanded from *Escherichia coli* and *Saccharomyces cerevisiae* to *Pichia pastoris* in the reconstitution of multi-gene biosynthetic pathways in order to improve expression levels and overproduce plant proteins [16]. In addition to increasing the yield, manufacturing plant alkaloids in heterologous systems enhances our ability to metabolically engineer beneficial compounds which we have not seen in nature with potentially improved biological activities.

Here, we provide an overview of recent advances and discuss current approaches, as well as potential future directions, in the metabolic engineering of MIAs. We also highlight the benefits of heterologous production and the engineering of MIAs in both microbial and plant systems, such as in tobacco plants. Engineering plant natural product pathways requires complete elucidation of the biosynthetic genes and regulatory mechanisms to optimize the yield [17]. Recent advances in decoding MIA biosynthetic pathways were made possible through the biochemical characterization of these enzymes in both native and engineered microbial and plant systems. Subsequently, the further fine-tuning of these biological components has led to the production of desired products at higher titers.

## 2. The Complexities of MIA Biosynthetic Pathway

Almost all MIAs are derived from the central precursor strictosidine. Upon removal of glucose on strictosidine by β-glucosidase, this important intermediate is extremely malleable for further conversion to various MIA backbones [1]. The production of strictosidine is significant because the reconstitution of biosynthetic pathways in generating this MIA core scaffold represents a critical first step for manufacturing high-value MIAs in microbial systems [18]. This metabolic engineering effort includes the production of the monoterpene precursor to form iridoids. The entire iridoid pathway from the 2-*C*-methyl-ᴅ-erythritol 4-phosphate (MEP) pathway has been elucidated. The first committed step of monoterpene synthesis in plants begins with the hydrolysis of geranyl pyrophosphate (GPP) by geraniol synthase (GES) [19]. Geraniol is then hydroxylated by the cytochrome P450, geraniol 8-hydroxylase (G8H) [20]. The next step includes the oxidation of 8-hydroxygeraniol to produce the dialdehyde 8-oxo-geranial by 8-hydroxygeraniol oxidoreductase (8HGO) and reductive cyclization to generate nepetalactol by iridoid synthase (ISY) [21,22]. An additional protein, known as the major latex protein-like (MLPL) enzyme, has been reported to prevent shunt product formation, thereby increasing the efficiency of nepetalactol production [23]. Nepetalactol undergoes oxidation by the cytochrome P450, iridoid oxidase (IO) to 7-deoxyloganetic acid, which is then glycosylated to generate 7-deoxyloganic acid, followed by hydroxylation to produce loganic acid and methylation to produce loganin by loganic acid *O*-methyltransferase (LAMT) [24,25,26,27,28]. The last step of the iridoid pathway involves an additional cytochrome P450, secologanin synthase (SLS), which catalyzes the oxidative ring opening to generate a reactive aldehyde, secologanin. Secologanin is then coupled to tryptamine by a Pictet-Spenglerase, called strictosidine synthase (STR) [29,30] (Figure 1).

Recently, multiple enzymes that contribute to the generation of various MIA backbones have been identified [13,14,15,31]. The branching points of this part of the MIA pathway are critical to ensuring the accumulation of the desired products. While the reactivity and instability of strictosidine aglycone are part of the challenges involved in the elucidation of these enzymes, the plasticity of the pathways is an advantage that allows for control of the direction in terms of generating various MIA skeletons, such as those observed in iboga (i.e., catharanthine), aspidosperma (i.e., tabersonine, vindoline), and corynanthe (i.e., ajmalicine, yohimbine, alstonine) alkaloids [13,14,31,32] (Figure 1).

The genes that link strictosidine aglycone to the formation of iboga MIA, catharanthine, and the aspidosperma MIA tabersonine, including geissoschizine synthase (GS), geissoschizine oxidase (GO), *O*-acetylstemmadenine oxidase (ASO), precondylocarpine acetate synthase (PAS), and dihydroprecondylocarpine synthase (DPAS), have been identified. They set a foundation on which to develop an understanding the branching points to produce vinblastine in *C. roseus* [13,14,15,31]. The anticancer drug vinblastine is the dimerization product of two MIAs, catharanthine and vindoline, catalyzed by a peroxidase [33]. While catharanthine is generated through nine enzymatic steps from strictosidine, vindoline is derived from tabersonine via seven steps, which have been fully elucidated [12,34,35,36,37,38] (Figure 1). The functional characterization of those key reactions to the formation of MIAs permits further development in terms of engineering the entire MIA pathways in heterologous systems for enhanced biotechnological applications.

Besides the efforts of dealing with a long biosynthetic pathway of 31 enzymatic steps, vinblastine biosynthesis is highly compartmentalized [39]. The assembly of loganic acid, starting from geranyl pyrophosphate (GPP), occurs in specialized cells of mesophyll called internal-phloem-associated parenchyma cells (IPAP) [19,24,25,26,27]. Loganic acid is then transported to the leaf epidermis, where it is methylated, followed by the oxidative ring opening to generate secologanin. The leaf epidermis is also the main site for condensation between secologanin and tryptamine, and is mediated by strictosidine synthase (STR) to form strictosidine, which is then deglycosylated by strictosidine β-D-glucosidase (SGD) [40]. The reconstitution of the MIA pathway is also made challenging by the compartmentalization of enzymes across at least five different subcellular compartments, including the cytoplasm, plastid for geraniol synthase [19], vacuole for strictosidine synthase (STR), nucleus for strictosidine β-ᴅ-glucosidase (SGD), and endoplasmic reticulum (ER) for multiple cytochrome P450s and their reductases [41,42]. The spatial and regulatory aspects of pathway reconstitution are thus critical to count for trafficking processes that may require active specific transporter proteins, particularly in microbial hosts.

The metabolic engineering of natural products in the native plant host is typically more burdensome, depending on the plant species, especially due to difficult gene transformation, particular preference of propagation conditions, and intricate genetic regulation. Homologous reconstitutions of MIA pathways have been reported in cell suspension and hairy root cultures with the goals of increased yield [43]. Although efficient transient expression and stable genetic transformation of *C. roseus* plants are feasible, genetic engineering of the native producer plant remains challenging. These efforts involve overexpressing the MIA biosynthetic genes and, possibly, other primary metabolic pathway genes that regulate various branches of MIA pathways, such as impacting the iridoid pathway [44]. Other examples of genetic manipulation include expressing constitutive biosynthetic enzymes, transcription factors, or transporters known to regulate MIA accumulation [43,45,46]. While most engineering efforts of the MIA pathways into microbial and plant systems are intended to increase the yield of the desired MIAs, a notable advantage of reconstitution of natural product pathways includes the expansion of chemical diversity, potentially enabling the production of other bioactive MIAs that are not readily available in nature [47,48,49].

## 3. Engineering of MIA Pathway in Microbial Hosts

With continuous enhancements in genomics and metabolomics approaches, metabolic engineering efforts are now rapidly growing, including exploring possible other host systems to continue increasing the yields of the end products [50]. The preferred systems are typically easier to cultivate and efficiently transformed with faster and more economical options. While *E. coli* and *S. cerevisiae* are the most common microbial hosts for heterologous expression of plant biosynthetic genes due to their quick generation times and the widespread availability of genetic editing tools, more microbial species, including *P. pastoris,* are now further utilized for biotechnological productions of plant natural products. Currently, *S. cerevisiae* (baker’s yeast) is the first choice for a eukaryotic system owing to its effectiveness in the functional expression of microsomal plant cytochrome P450s (CYPs). Because of CYPs’ involvement in oxidation reaction, catalysis is coupled to the reduction in NADPH, which requires a cytochrome P450 reductase (CPR) partner to shuttle electrons, while other proteins, such as cytochrome b5 and cytochrome b5 reductase, may be added to the system [51,52]. Moreover, the effects of extensive intracellular compartmentalization, such as in MIA biosynthetic pathways, can only be studied further in a eukaryotic host [42,43,52]. The applications of *S. cerevisiae* for gene discovery and identification of missing pathways are now adopted more often [12,13,31,53].

The rapid generation time of engineered yeast with microbial fermentation offers great advantages, including reduced time, space, and resources that are typically required for plant propagation and metabolite extraction [54]. Additional benefits also include the consistency during batch-to-batch industrial manufacturing process and the ability to achieve high purity levels of the targeted metabolites. The applications of yeast as a microbial host for the synthesis of complex non-MIA natural products have been described for the production of artemisinin [51,55], tropane alkaloid [53], opioid thebaine and hydrocodone [56], and noscapine [57].

The first reported effort to express an MIA biosynthetic gene in *S. cerevisiae* was heterologous strictosidine production through feeding secologanin and tryptamine by strictosidine synthase (STR) [58]. Later, as the entire iridoid pathway was elucidated, the first de novo strictosidine biosynthesis in *S. cerevisiae* was achieved despite a low titer of 0.5 mg/L [18]. As more MIA biosynthetic genes were characterized with the involvement of unstable intermediates, expressing multiple genes of the same pathway became more appealing, as demonstrated in the biochemical characterization of the missing genes of the vindoline pathway: tabersonine 3-oxygenase (T3O) and tabersonine 3-reductase (T3R) [12]. Although the primary goal of this study is to functionally characterize the targeted genes, this was a remarkable achievement in terms of producing vindoline, the precursor of vinblastine in yeast via a seven-step pathway from tabersonine (Figure 1) [12]. Further strain optimization using several strategies was later considered for the further development of metabolic engineering in microbial systems, which relied on the unique characteristics of MIA biosynthetic pathways [59].

Throughout the process of developing the *S. cerevisiae* strain in order for it to produce strictosidine, various approaches were reported. Brown et al. successfully integrated fourteen MIA biosynthetic genes, with an additional seven accessory genes and three gene deletions into the yeast genome [18,60]. This study provides a foundation for the metabolic engineering of *S. cerevisiae* in order to establish a microbial cell factory for the future production of complex MIAs. Another recent example also emerged to optimize the early biosynthetic steps of the MIA pathway, particularly to produce monoterpene nepetalactol [61]. Campbell and coworkers successfully produced monoterpenes in yeast by overexpression of four enzymes in the mevalonate pathway of yeast, as well as integrated geraniol synthase from *Ocimum basilicum* and geranyl diphosphate synthase from *Abies grandis* [61]. This study demonstrated the need to exploring homologous genes from different organisms as part of the engineering scheme for the overproduction of targeted metabolites. Interestingly, Yee and coworkers effectively engineered *S. cerevisiae* by targeting the geraniol biosynthetic pathway to the mitochondria to keep the GPP pool from using the cytosolic ergosterol pathway [62]. Their efforts resulted in fed-batch fermentation reaching a titer of 227 mg/L when producing 8-hydroxygeraniol, and they further integrated additional iridoid biosynthetic genes to generate nepetalactol [62].

Further engineering of *S. cerevisiae* to overproduce strictosidine improved significantly with additional knowledge of shunt pathways in iridoid biosynthesis [23,63]. Billingsley et al. included metabolic flux and interference analysis, which demonstrated the two critical steps needed to increase the levels of iridoids in *S. cerevisiae*. The oxidation of 8-hydroxygeraniol to 8-oxogeranial and NADPH-dependent reductive cyclization forming nepetalactol were the keys for improved biocatalytic capabilities. They also reported the significance of deleting five genes involved in α,β-unsaturated carbonyl metabolism to achieve higher levels of the targeted iridoid with a reduced quantity of shunt products [63]. Further efforts for the higher-titer production of strictosidine (~50 mg/L) continued as the promoter system that influences yeast growth and metabolite production was optimized [47]. This was achieved by tuning the expression of CYPs and their accessory enzymes, as well as adding the MLPL enzyme to reduce shunt product formation. In addition, Misa and colleagues reported the production of halogenated strictosidine analogs upon feeding 7-fluorotryptamine and 7-chlorotryptamine [47].

The higher production of vindoline in *S. cerevisiae* was also achieved after Qu et al. [12] successfully assembled a seven-step pathway by employing CRISPR (Clustered Regularly Interspaced Short Palindromic Repeats)/Cas9-mediated multiple genome integration technology [12,64]. By increasing the copy numbers of the vindoline pathway genes, pairing CYPs with compatible CPRs, and improving the cofactor supply and fermentation conditions, they reached a final titer of ~16.5 mg/L of vindoline [64]. In another study, tabersonine-to-vindoline conversion was improved by a rational medium optimization and sequential feeding approach while modifying the heterologous gene copies in order to restrain the production of side metabolites in the pathway [65]. Altogether, these studies further highlight the success of using the microbial cells to overproduce medicinally important compounds.

Recently, the de novo biosynthesis of catharanthine and vindoline was accomplished in the yeast cell factory. This effort is an important milestone in the engineering of plant natural products in a heterologous system, as MIA biosynthetic pathways are considerably long, with multiple branching points (Figure 1) [66,67]. In this study, Zhang and coworkers performed 56 genetic edits, involving the integration of 34 biosynthetic genes, and divided the *C. roseus* pathway into three parts: (1) a strictosidine module expressing iridoid biosynthetic genes; (2) a catharanthine/tabersonine module expressing the strictosidine synthase (STR), SGD, multiple oxidoreductases, and hydrolases to generate catharanthine and tabersonine; and (3) a vinblastine module including vindoline biosynthetic genes and peroxidase 1 (PRX1) [14,15,18,33,66]. In addition, each module incorporated the *C. roseus* cytochrome P450 reductase (CPR) and cytochrome b5 (CYB5) to sustain the cytochrome P450 enzymes [66,67]. As part of the effort to overcome the bottlenecks in generating strictosidine, they also utilized full-length STRs, integrated *Vinca minor* 8HGO for the iridoid pathway, and incorporated the active domain of SGD from their SGD hybrid development [66]. In their efforts to engineer each module, the researchers tested the functionality and efficiency of each catalytic step by feeding the exogenous substrate, as well as product measurement [67]. While Zhang et al. produced catharanthine and vindoline at titers of 91.4 µg/L and 13.2 µg/L, respectively, higher titers were reported by Gao et al. [68], reaching titers of 527.1 µg/L of catharanthine and 305.1 µg/L of vindoline. In this study, Gao and colleagues highly utilized the CRISPR/Cas9 system to modify the *S. cerevisiae* genome, while they also divided the complex MIA pathways into three parts to generate strictosidine aglycone, catharanthine/tabersonine, and vindoline [68]. Previously, their research group also constructed a *S. cerevisiae* strain that overproduced ajmalicine [69] (Figure 1). Their success in achieving 61.4 mg/L of ajmalicine by optimizing the iridoid pathway, leading to the formation of strictosidine aglycone, was another fundamental groundbreaking metabolic engineering effort using the CRISPR/Cas9 genome editing system [69]. By ensuring the necessary supply of various cofactors and alleviating the bottlenecks of the rate-limiting steps, they effectively engineered a yeast strain with remarkably efficient machinery.

While model systems like *S. cerevisiae* are commonly preferred hosts for the reconstitution of plant natural product pathways, another prospective chassis for biotechnology application has been explored. The methylotrophic yeast *P. pastoris* (*Komagataella phaffii*) is considered one of the most popular protein expression systems and a safe host used in the pharmaceutical industry; nevertheless, its homologous recombination machinery is inefficient in contrast to *S. cerevisiae*. Recently, the emergence of CRISPR-Cas9 technologies has begun to offer new avenues for highly efficient one-step genomic multi-loci integration of genes in *P. pastoris* [70,71]. The methods and tools, such as gene expression cassettes for the simultaneous integration of multiple genes in engineering *P. pastoris,* are more readily available [16,70,72]. Remarkably, the de novo biosynthesis of catharanthine in *P. pastoris* was recently described, allowing for improved production of catharanthine from simple carbon sources at a maximum titer of 2.57 mg/L in comparison to previously reported yield from *S. cerevisiae* [73,74]. This study highlights the significance of optimization strategies developed for reconstitution, such as the complex long MIA pathway in *P. pastoris* leading to catharanthine, by addressing various approaches. This includes the selection of suitable plant enzymes with enhanced functionality, facilitating the trafficking of substrates by expressing fusion protein, and the amplification of genes involved in rate-limiting steps [73,74]. The rapid development of tools in synthetic biology, including the application of CRISPR/Cas9, boosts the prospect of overcoming challenges in metabolic regulation to improve precursor supplies and ensures balanced microbial cell growth in the future industrial production of high-value natural products. Overall, the reconstitution of catharanthine and vindoline biosynthetic pathways in *S. cerevisiae* and *P. pastoris*, although vinblastine still needs to be produced semi-synthetically, demonstrate huge prospects in utilizing microbial systems to cost-effectively manufacture anticancer MIAs.

## 4. Engineering of the MIA Pathway in Heterologous Plant Hosts

Creating a knock-out line or overexpression of plant natural product biosynthesis is a common approach to studying a particular reaction for the purpose of validating the function of a target protein [17]. In addition, more advanced techniques have been developed for highly efficient plant genetic transformation. Currently, *Nicotiana benthamiana* is the most attractive platform of efficiently transformable model plants, especially in efforts to reconstitute natural product pathways. Several classes of plant natural products have been successfully manufactured in *N. benthamiana*, such as lignan derived etoposide [75], betalains [76], and polyketides [77]. Due to the remarkable versatility of the metabolism, matched with the compartmentalization necessary for the reconstitution of functional pathways, the short life cycle, and the amenability to Agrobacterium-mediated transient expression, *N. benthamiana* is an ideal chassis for manufacturing specialized plant metabolites, including MIAs [78,79]. This well-established heterologous system provides additional flexibility in the expression of plant biosynthetic genes from other species, therefore creating potential for manufacturing high-value compounds on an industrial scale. Various methods for the de novo synthesis of plant natural product pathways have also been developed. They include infiltration with a strain of *Agrobacterium tumefaciens* harboring the gene of interest in a plant expression vector by using syringe or vacuum infiltration, virus-mediated overexpression, gene silencing, gene editing, or nuclear/plastid transformation [50,54,78,79]. Considering that subcellular compartmentalization in monoterpene biosynthesis relies on GPP availability in plastids, cytosol, or mitochondria, *N. benthamiana* is suitable for the reconstitution of complex MIA biosynthetic pathways, possibly by enriching the pools of MIA precursors [80].

The first effort to integrate an MIA pathway in *N. benthamiana* successfully produced strictosidine in seven steps by supplying intermediate iridodial while functionally characterizing multiple iridoid candidate genes [25]. The suitability of *N. benthamiana* as a model plant host for the increased production of MIAs was further confirmed by Dudley and colleagues, particularly in fine-tuning the MIA biosynthetic pathway to increase the yield of strictosidine [81]. To increase the flux of isoprenoid precursors, they divided the early pathway for chloroplast localization, while the later steps were localized to the cytosol to imitate the subcellular compartmentalization of *C. roseus*. Furthermore, the activity of endogenous enzymes of *N. benthamiana* was evaluated, particularly the roles of oxidases in generating oxidized derivatives of geraniol and glycosyltransferases that could result in the accumulation of unintended side products [81,82]. While the removal of endogenous glycosyltransferases did not influence the yield of strictosidine, they demonstrated the crucial role of the MLPL protein in directing the stereoselectivity of ring closure in iridoid synthesis for the purpose of improving flux through the pathway, therefore maximizing strictosidine production [81].

Recently, the late-stage MIA pathway from strictosidine to produce catharanthine and tabersonine was successfully reconstituted in *N. benthamiana* using a modular Agrobacterium-mediated transient gene expression approach [83]. In this study, Grzech and colleagues optimized the yields of the late vinblastine precursor, precondylocarpine acetate, to reach as high as ~2.7 mg per 1 g of frozen tissue [83]. Their successful reconstitution of the MIA pathway in *N. benthamiana* demonstrated that a plant-based system can be highly promising, even for the integration of complex pathways to manufacture high-value natural products. Interestingly, besides improving the yield of the desired compound in heterologous plant hosts, transient expression in Nicotiana plants facilitates efficient generation of molecular diversity. For instance, Boccia and colleagues utilized unnatural starting substrates to generate MIAs that do not commonly occur in nature [49]. The derivatization of natural products in heterologous plant systems lays a substantial foundation for further expansion of metabolic engineering in the development of novel compounds with improved pharmacological properties, or for enhancing the biological activity of the parent compound. Overall, advanced strategies in engineering plant systems have accelerated the optimization of a favored host to control the expression of plant natural products’ biosynthetic pathways to achieve increased production of high-value compounds.

## 5. Conclusions and Future Directions

Utilization of heterologous host production systems for MIA biosynthesis has been proven to have immense potential to inspire further reconstitution of complex plant natural product pathways. While microbial production systems with fermentation-based approaches may support the development of robust manufacturing platforms, heterologous plant systems offer the advantages of compatibility in the regulation of plant metabolism and subcellular compartmentalization [50]. The progress of metabolic engineering efforts for the production of MIAs in both microbial and plant systems is summarized in Table 1. Overall, the rapidly advancing strategies to engineer heterologous hosts could lower the cost of manufacturing these high-value natural products [17]. The demonstrations of de novo catharanthine and vindoline production discussed in this review emphasize significant milestones in a new era of advanced genome editing technology, which will ultimately secure our access to more stable anticancer vinblastine supply chains. While the de novo biosynthesis of MIAs could be further optimized by refining the host genes, the engineered heterologous systems offer the additional advantages of generating new-to-nature MIAs that could be tested for their biological activities. Further engineering efforts may require altering subcellular localizations and investigating post-translational modifications by signal peptide substitution or truncation. Replacing certain catalytic enzymes with homologous proteins has also been an effective approach, and could also expand the application of directed evolution, especially for biosynthetic enzymes that can be further engineered. In the future, the engineering of biosynthetic enzymes with synthetic biological approaches will result in increased prosperity in terms of generating compounds that have not been observed in nature, especially with new structural MIA scaffolds, thereby further advancing drug discovery.

## Figures and Tables

**Figure 1 biology-12-01056-f001:**
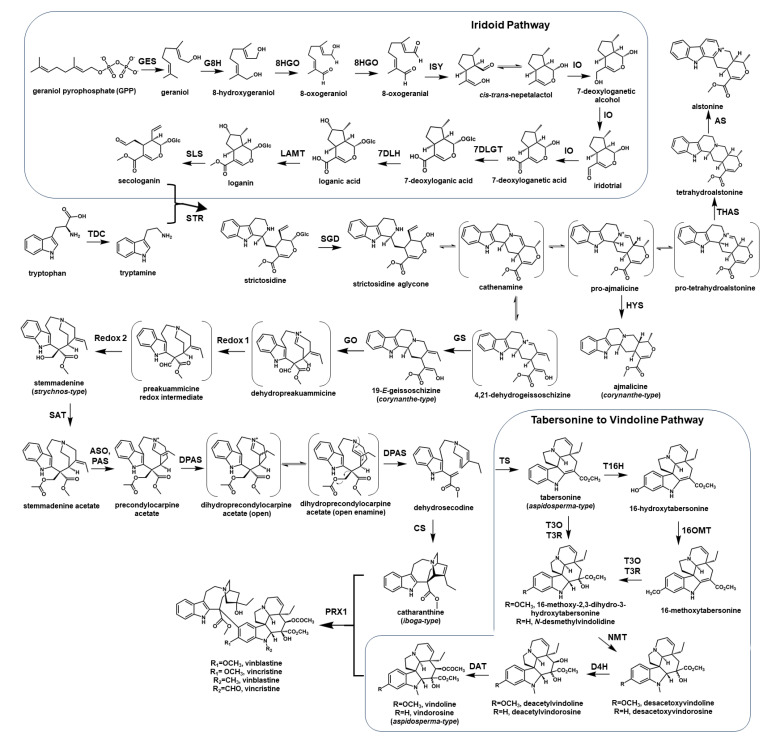
The monoterpene indole alkaloid (MIA) biosynthetic pathways in *Catharanthus roseus* to produce different types of MIAs (iboga, aspidosperma, corynanthe-types), including the formation of vinblastine and vincristine. GES, geraniol synthase; 8GH, geraniol 8-hydroxylase; 8HGO, 8-hydroxygeraniol oxidoreductase; ISY, iridoid synthase; IO, iridoid oxidase/7-deoxyloganetic acid synthase; 7DLGT, 7-deoxyloganetic acid glucosyltransferase; 7DLH, 7-deoxyloganic acid hydroxylase; LAMT, loganic acid *O*-methyltransferase; SLS, secologanin synthase; TDC, tryptophan decarboxylase; STR, strictosidine synthase; SGD (strictosidine *O*-β-D-glucosidase); THAS, tetrahydroalstonine synthase; AS, alstonine synthase; HYS, heteroyohimbine synthase; GS, geissoschizine synthase; GO, geissoschizine oxidase; Redox 1, oxidation-reduction reaction 1; Redox 2, oxidation-reduction reaction 2; SAT, stemmadenine acetyltransferase; ASO, *O*-acetylstemmadenine oxidase; PAS, precondylocarpine acetate synthase; DPAS, dihydroprecondylocarpine acetate synthase; CS, catharanthine synthase/hydrolase 1; TS, tabersonine synthase/hydrolase 2; T16H, tabersonine 16-hydroxylase; 16OMT, 16-*O*-methyltransferase; T3O, tabersonine 3-oxidase; T3R, tabersonine 3-reductase; NMT, *N*-methyltransferase; D4H, desacetoxyvindoline 4-hydroxylase; DAT, deacetylvindoline acetyltransferase; PRX1, peroxidase 1.

**Table 1 biology-12-01056-t001:** Summary of engineering efforts in the reconstitution of monoterpene indole alkaloid (MIA) pathways in heterologous systems.

Host Species	Targeted Metabolite	Key Steps of Engineering	Yield	Ref
DE NOVO PRODUCTION IN MICROBIAL SYSTEMS				
*Saccharomyces cerevisiae*	Strictosidine	Incorporation of iridoid biosynthetic genes, 15 plant-derived genes, 1 avian gene, 5 yeast genes, 3 gene deletions	0.5 mg/L strictosidine from optimized pathway Detection of 0.8 mg/L loganin	[18]
*Saccharomyces cerevisiae*	Nepetalactol	Mevalonate pathway optimization overexpression genes, integration of monoterpene biosynthetic genes	11.4 mg/L geraniol 5.3 mg/L 8-hydroxygeraniol No significant yield of nepetalactol reported	[61]
*Saccharomyces cerevisiae*	Nepetalactol	Iridoid pathway optimization, targeting the geraniol biosynthetic pathway to the mitochondria	7 mg/L geraniol 227 mg/L 8-hydroxygeraniol from geraniol, 5.9 mg/L of nepetalactol (11% conversion from 8-hydroxygeraniol)	[62]
*Saccharomyces cerevisiae*	Ajmalicine	Stable integration in the genome, construction of ajmalicine (29 expression cassettes) and sanguinarine (24 expression cassettes), multiplex genome integration	119.2 mg/L heteroyohimbine alkaloids containing 61.4 mg/L ajmalicine	[69]
*Saccharomyces cerevisiae*	Catharanthine, vindoline, vinblastine	Optimization of iridoid pathway to produce strictosidine, testing the full length of STR, testing hybrids of SGD domains, incorporating extra copies of vindoline biosynthetic genes	0.0132 mg/L vindoline 0.0914 mg/L catharanthine Semi-synthesis of vinblastine	[66]
*Saccharomyces cerevisiae*	Catharanthine, vindoline	Optimization of vindoline and catharanthine biosynthetic pathways, incorporating extra copies of rate-limiting enzymes	0.527 mg/L catharanthine 0.305 mg/L vindoline	[68]
*Pichia pastoris*	Catharanthine	CRISPR/Cas9-based genome integration; optimization of de novo biosynthesis of nepetalactol, strictosidine, and catharanthine pathways	2.57 mg/L catharanthine	[74]
FEEDING WITH PRECURSORS IN MICROBIAL SYSTEMS				
*Saccharomyces cerevisiae*	Vindoline	Vindoline biosynthetic genes from tabersonine, incorporation of 8 genes including all vindoline pathway genes and Cytochrome P450 Reductase (CPR)	1.1 mg/L vindoline from 17 mg/L of tabersonine	[12]
*Saccharomyces cerevisiae*	Nepetalactol	Deletion of old yellow enzymes (OYEs), synthesis of iridoid scaffold	45 mg/L iridoid (8-oxogeraniol, nepetalactol detected without specific yield for each iridoid)	[63]
*Saccharomyces cerevisiae*	Vindoline	Increase flux of vindoline pathway	266 mg/L vindoline (88% yield) from tabersonine 4.7 mg/L vindorosine from tabersonine	[65]
*Saccharomyces cerevisiae*	Vindoline	Integration of multiple copies of vindoline biosynthetic genes, CRISPR/Cas9 mediated multiplex integration technology to increase yield of vindoline	16.5 mg/L vindoline converted from total of 100 mg/L of tabersonine	[64]
*Saccharomyces cerevisiae*	Strictosidine	Iridoid pathway optimization, feeding modified tryptamine for the synthesis of halogenated strictosidine	56.2 mg/L strictosidine from 336.5 mg/L of nepetalactol and 320.4 mg/L of tryptamine	[47]
*Saccharomyces cerevisiae*	Catharanthine, tabersonine	Signal peptide modification of *O*-acetylstemmadenine oxidase to improve protein *N*-glycosylation, feeding with strictosidine analogs to produce fluorinated and hydroxylated catharanthine and tabersonine	0.021 mg/L of catharanthine from 15 mg/L of strictosidine aglycone, 0.128 mg/L of catharanthine from 15 mg/L of 19*E*-geissoschizine	[48]
HETEROLOGOUS PLANT SYSTEMS				
*Nicotiana benthamiana*	Strictosidine	Functional characterization of iridoid biosynthetic genes, incorporation of multiple iridoid pathway genes in heterologous plant systems	No yield reported	[25]
*Nicotiana benthamiana*	Strictosidine	De novo production of strictosidine, expression of 14 enzymes, amplification of iridoid pathway genes, incorporation of the major latex protein-like (MLPL) enzyme to improve the flux of the iridoid pathway	0.23 mg strictosidine per g dry weight	[81]
*Nicotiana benthamiana*	Precondylocarpine acetate, catharanthine, tabersonine	Use of modular vector assembly system for the optimization of six biosynthetic steps to produce precondylocarpine acetate from strictosidine, and reconstitution of catharanthine and tabersonine pathways	2.7 mg precondylocarpine acetate, 60 ng catharanthine, 10 ng tabersonine per g frozen tissue, leaves were infiltrated with 1 mL of 200 µM strictosidine	[83]
*Nicotiana benthamiana*	Alstonine, stemmadenine analogs	Chemical derivatization of MIAs by incorporating six individual biosynthetic pathway genes from strictosidine to alstonine and stemmadenine acetate	28 ng of 4-fluoroalstonine per g of plant fresh weight, 155 ng of 5-fluoroalstonine per g of plant fresh weight, 190 ng of 6-fluoroalstonine per g of plant fresh weight, 122 ng of 7-fluoroalstonine per g of plant fresh weight, 27 ng of 7-chloroalstonine per g of plant fresh weight when plants were infiltrated with each strictosidine analog, 25 ng of alstonine per g of plant fresh weight when plants were infiltrated with natural strictosidine (2 mL per leaf of 200 µM of the desired strictosidine analog)	[49]

## Data Availability

Not applicable.

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
