# Peer review of "Advances in Metabolic Engineering of Plant Monoterpene Indole Alkaloids"

_biology, 2023, doi:10.3390/biology12081056_

Round 1

Reviewer 1 Report

The review by Salim et al. describes recent efforts in the production of monoterpene indole alkaloids (MIAs) in yeast and Nicotiana benthamiana by metabolic engineering. MIAs are a large class of important bioactive metabolites and drug precursors, and producing these compounds by metabolic engineering is a timely and important topic. In my opinion, this review provides a comprehensive overview over current and recent achievements in this field that has not been covered in other recent reviews. It is therefore principally suitable for publication in Biology, in my opinion. However, several aspects need to be improved before this manuscript can be accepted. Most importantly, Figure 1 contains many mistakes and missing stereogenic information in the structures as well as typos that have to be corrected.

Figure 1: please check all structures very carefully and correct

  • Nepetalactol to secologanin: missing stereochemistry at bridgehead carbon
  • Strictosidine: 3alpha and S are redundant stereochemical identifiers - one or both could be omitted for clarity
  • Strictosidine aglycon: Missing H at C-3; missing O at C-21 (no double bond); missing stereochemistry at C-15 and C-20
  • Strictosidine to Geissoschizine: The bond at COOCH3 to C-16 should not be at the CH3 group
  • Cathenamine: Missing oxygen next to C-21
  • 19-E-geissoschizine: E should be in italics
  • Precondylocarpine/dihydroprecondylocarpine acetate: the position of the C=N double bond is incorrect, it has to be next to the ethylidene/ethyl side chain; also missing "o" precondyl[o]carpine
  • Tabersonine, vindoline, vincristine, vinblastine: missing stereochemistry at C-21
  • 16-Methoxy-2,3-dihydro-3-hydroxytabersonine: The complete upper part of the molecule is missing; also missing stereochemistry at C-21 and C-2
  • Vincristine and vinblastine: The catharanthine-derived subunit is incorrectly linked - correct carbons with five bonds
  • Figure legend: 7-deoxyloganic acid hydrolase: should be hydro[xy]lase
  • 8-HGO: use consistent name; in the main text you previously used (the more common name) oxidoreductase instead of oxidase
  • IS: should be called ISY, iridoid synthase and not IS, iridodial synthase
  • IO: should be called iridoid oxidase, not iridodial oxidase

Table 1:

  • In my opinion, this table would benefit from clarifying which examples represent complete de novo production and which start at later stages from fed precursors (and from which intermediate). This is very important information and I think it should be highlighted better. Without clarifying this, many of the examples are hardly comparable. For example, ref. 61 is not de novo production and the yields are therefore strongly misleading when comparing to ref. 12 or 69.
  • Entry regarding ref. 58: What do you mean by iridoid? To which compounds does the yield refer?

Other suggestions:

  • l. 121 "biosynthesis is highly compartmentalized": The most recent publication for this aspect should be cited: https://www.nature.com/articles/s41589-023-01327-0
  • l. 199/200 "Campbell and coworkers …" : please explain the relevance of citronellol in this context or rephrase. Yes, it's a monoterpene, but it's not en route to MIAs. It's not clear to me what the relevance of this statement is.
  • l. 230 "final titer" of which compound?

Minor typos/language that should be fixed:

  • l. 79: 8-hydroxy[l]geraniol remove "l"
  • l. 109: genes that link[s] remove "s"
  • l. 111/112: precondyl[o]carpine
  • l. 114: "anticancer vinblastine" anticancer drug?
  • l. 134: "… aspects of pathway reconstitution is thus critical" should be "are"
  • l. 161: "is now further utilized" should be "are"?
  • l. 296: "ketides" should be "polyketides"
  • Table 1: "10-hydroxygeraniol" is now more commonly called 8-hydroxygeraniol

Author Response

Thank you for your feedback and comments on the significance of this manuscript.

In response to your comments on Figure 1, we have corrected each aspect mentioned and provided additional important intermediates and improved labeling. All structures have been carefully inspected.

  • Nepetalactol to secologanin: missing stereochemistry at bridgehead carbon It has been corrected.
  • Strictosidine: 3alpha and S are redundant stereochemical identifiers - one or both could be omitted for clarity It has been corrected to strictosidine.
  • Strictosidine aglycon: Missing H at C-3; missing O at C-21 (no double bond); missing stereochemistry at C-15 and C-20 it has been corrected.
  • Strictosidine to Geissoschizine: The bond at COOCH3 to C-16 should not be at the CH3 group it has been corrected.
  • Cathenamine: Missing oxygen next to C-21 It has been corrected.
  • 19-E-geissoschizine: E should be in italics It has been corrected.
  • Precondylocarpine/dihydroprecondylocarpine acetate: the position of the C=N double bond is incorrect, it has to be next to the ethylidene/ethyl side chain; also missing "o" precondyl[o]carpine. - It has been corrected.
  • Tabersonine, vindoline, vincristine, vinblastine: missing stereochemistry at C-21- It has been corrected.
  • 16-Methoxy-2,3-dihydro-3-hydroxytabersonine: The complete upper part of the molecule is missing; also missing stereochemistry at C-21 and C-2 - It has been corrected.
  • Vincristine and vinblastine: The catharanthine-derived subunit is incorrectly linked - correct carbons with five bonds - It has been corrected.
  • Figure legend: 7-deoxyloganic acid hydrolase: should be hydro[xy]lase - It has been corrected.
  • 8-HGO: use consistent name; in the main text you previously used (the more common name) oxidoreductase instead of oxidase - It has been corrected to oxidoreductase.
  • IS: should be called ISY, iridoid synthase and not IS, iridodial synthase It has been corrected to iridoid synthase.
  • IO: should be called iridoid oxidase, not iridodial oxidase It has been corrected to iridoid oxidase.

In response to your feedback on Table 1,

  • „In my opinion, this table would benefit from clarifying which examples represent complete de novo production and which start at later stages from fed precursors (and from which intermediate). This is very important information and I think it should be highlighted better. Without clarifying this, many of the examples are hardly comparable. For example, ref. 61 is not de novo production and the yields are therefore strongly misleading when comparing to ref. 12 or 69.“ - We have revised the content by differentiating de novo productions from the ones that involve feeding precursors, as well as clarifying from which intermediate.
  • Entry regarding ref. 58: What do you mean by iridoid? To which compounds does the yield refer? The names of iridoid detected have been added to this table, although the authors of this research article did not specify the yield of each iridoid. This information has also been included in the table.

Other suggestions:

  • l. 121 "biosynthesis is highly compartmentalized": The most recent publication for this aspect should be cited: https://www.nature.com/articles/s41589-023-01327-0

-This publication has been added.

  • l. 199/200 "Campbell and coworkers …" : please explain the relevance of citronellol in this context or rephrase. Yes, it's a monoterpene, but it's not en route to MIAs. It's not clear to me what the relevance of this statement is. - Thank you for your suggestion. We agree that citronellol is not en route to MIAs, therefore we removed citronellol and focused on iridoids that are parts of the MIA pathways.
  • l. 230 "final titer" of which compound? We have added the final titer of vindoline.

Thank you for your suggestions. The following typos/language issues have been corrected.

Minor typos/language that should be fixed:

  • l. 79: 8-hydroxy[l]geraniol remove "l" It has been corrected.
  • l. 109: genes that link[s] remove "s" It has been corrected.
  • l. 111/112: precondyl[o]carpine It has been corrected.
  • l. 114: "anticancer vinblastine" anticancer drug? It has been added.
  • l. 134: "… aspects of pathway reconstitution is thus critical" should be "are" It has been corrected.
  • l. 161: "is now further utilized" should be "are"? It has been corrected.
  • l. 296: "ketides" should be "polyketides" It has been corrected.
  • Table 1: "10-hydroxygeraniol" is now more commonly called 8-hydroxygeraniol It has been revised to 8-hydroxygeraniol.

Reviewer 2 Report

The manuscript summarized recent (5 years) researches on metabolic engineering of monoterpenoid Indole alkaloid in microbial and plant systems, and provided a quick reference point for readers not in the field. Some minor changes are suggested:

Line 52: …expanded from E. coli and Saccharomyces cerevisiae to include …

Line 62: …the biosynthetic genes and regulatory mechanism to optimize the yield…

Line 88: …secologanin. Secologanin is then coupled to tryptamine…

Figure 1: the two enzymes Redox 1, Redox 2 after GO are missing in the pathway.

The current figure suggests that vincristine and vinblastine are made from different pathway. Consider redrawing the figure for clarity. 

The DPAS reaction includes deacetylation, consider redrawing the intermediate

Line 153 should include the reference Shahsavarani et al. 2023 Metabolic Engineering Communication (Improved protein glycosylation enabled heterologous biosynthesis of monoterpenoid indole alkaloids and their unnatural derivatives in yeast). The article shows the improvement of titers by improving protein glycosylation, and demonstrates the production of halogenated MIAs in yeast.

Line 170 …are more adopted nowadays… or other similar expression

The article is clear and concise, but could use a thorough check on syntax etc. 

Author Response

Thank you for your feedback and comments on the significance of this manuscript.

The following phrases have been revised:

Line 52: …expanded from E. coli and Saccharomyces cerevisiae to include …

Line 62: …the biosynthetic genes and regulatory mechanism to optimize the yield…

Line 88: …secologanin. Secologanin is then coupled to tryptamine…

Figure 1: the two enzymes Redox 1, Redox 2 after GO are missing in the pathway.

 The current figure suggests that vincristine and vinblastine are made from different pathway. Consider redrawing the figure for clarity. 

In response to your comments on Figure 1, we have corrected each aspect mentioned and provided additional important intermediates and improved labeling. All structures have been carefully inspected.

 The DPAS reaction includes deacetylation, consider redrawing the intermediate The precondylocarpine acetate intermediate has been included in this figure.

Line 153 should include the reference Shahsavarani et al. 2023 Metabolic Engineering Communication (Improved protein glycosylation enabled heterologous biosynthesis of monoterpenoid indole alkaloids and their unnatural derivatives in yeast). The article shows the improvement of titers by improving protein glycosylation, and demonstrates the production of halogenated MIAs in yeast. This publication has been cited and included in Table 1.

Line 170 …are more adopted nowadays… or other similar expression – It has been corrected.

Reviewer 3 Report

Monoterpene indole alkaloids (MIAs) encompass a diverse family of over 3000 plant natural products with a wide range of medical applications, and the famous MIAs are anticancer drugs vinblastine and vincristine. The low level of abundance of MIAs in natural sources hampered the utilization of MIAs, and synthetic biology is demonstrated to be a very useful approach to resolve this problem. The authors reviewed recent advances in metabolic engineering of MIAs, with focus on engineering microbial and heterologous plant systems in generating MIAs, and discussed the current issues and the future developments of manufacturing MIAs by synthetic biology approaches. The manuscript was written well, and there are only some minor mistakes.

(1)   Line 69: beta should be corrected to β, same with other writing in this manuscript

(2)   Line 87: generating was suggested to be replaced by “to generate”.

(3)   Line 93: Two enzymes Redox1 and 2 are not seen in Figure 1. Please check all abbreviations in the note of Figure 1, and make sure they are same with those in Figure 1.

(4)   Line 223: where halogen derived from? Please make it clear.

(5)   Line 235, 302,354: De novo should be written in italic style.

(6)   Line 342: Regarding MIAs biosynthesis is the longest plant natural product biosynthesis pathway ever manufactured in microorganism, I would see more intensive evaluation and discussions on the difficulties, challenges, in particular strategies in this aspects.

(7)   References: The styles of some references are not unified, please check them.

(8)   Other minor mistakes are highlighted in yellow color in the attached PDF file.

Author Response

Thank you for your feedback and comments on the significance of this manuscript. We have corrected the following aspects:

(1)   Line 69: beta should be corrected to β, same with other writing in this manuscript It has been corrected.

(2)   Line 87: generating was suggested to be replaced by “to generate”. It has been corrected.

(3)   Line 93: Two enzymes Redox1 and 2 are not seen in Figure 1. Please check all abbreviations in the note of Figure 1, and make sure they are same with those in Figure 1. -We have checked all abbreviations on the figure description to match with those in Figure 1 and the main text.

(4)   Line 223: where halogen derived from? Please make it clear. -The names of tryptamine analogs involved have been added: 7-fluorotryptamine and 7-chlorotryptamine.

(5)   Line 235, 302,354: De novo should be written in italic style. They have been corrected.

(6)   Line 342: Regarding MIAs biosynthesis is the longest plant natural product biosynthesis pathway ever manufactured in microorganism, I would see more intensive evaluation and discussions on the difficulties, challenges, in particular strategies in this aspects. This sentence has been revised to clarify the significance of complex MIA pathway reconstitution in microbial systems.

(7)   References: The styles of some references are not unified, please check them. -The references have been checked and revised.

Round 2

Reviewer 1 Report

I thank the authors for their corrections. I find the manuscript much improved, particularly Table 1. However, Figure 1 still contains some minor mistakes in the structures which need to be corrected before this manuscript is accepted.

  • Secologanin: still missing stereochemistry where the C2 aldehyde side chain is located
  • Strictosidine aglycone: the left structure is missing the oxygen at C17 - should be hydroxy group instead of methyl group
  • Cathenamine and the right strictosidine aglycone isomer are identical - why show both?
  • Alstonine: Missing stereochemistry at C15
  • 4,21-Dehydrogeissoschizine is identical to the left strictosidine aglycone isomer shown (except the missing oxygen) - why show both?
  • 4,21-Dehydrogeissoschizine: missing stereochemistry at C15
  • Most final pathway products lack stereochemistry at C2

Author Response

We highly appreciate the feedback from Reviewer 1 regarding Figure 1. The following mistakes have been corrected:

  • Secologanin: still missing stereochemistry where the C2 aldehyde side chain is located – It has been corrected.
  • Strictosidine aglycone: the left structure is missing the oxygen at C17 - should be hydroxy group instead of methyl group – It has been corrected and we have removed the redundancy.
  • Cathenamine and the right strictosidine aglycone isomer are identical - why show both? - We agreed with the identical structures shown previously. We have removed the redundancy.
  • Alstonine: Missing stereochemistry at C15 – It has been corrected.
  • 4,21-Dehydrogeissoschizine is identical to the left strictosidine aglycone isomer shown (except the missing oxygen) - why show both? - It has been corrected and we have removed the redundancy.
  • 4,21-Dehydrogeissoschizine: missing stereochemistry at C15 – It has been corrected.
  • Most final pathway products lack stereochemistry at C2 – It has been corrected.

Thank you.